# HNMT Upregulation Induces Cancer Stem Cell Formation and Confers Protection against Oxidative Stress through Interaction with HER2 in Non-Small-Cell Lung Cancer

**DOI:** 10.3390/ijms23031663

**Published:** 2022-01-31

**Authors:** Kuang-Tai Kuo, Cheng-Hsin Lin, Chun-Hua Wang, Narpati Wesa Pikatan, Vijesh Kumar Yadav, Iat-Hang Fong, Chi-Tai Yeh, Wei-Hwa Lee, Wen-Chien Huang

**Affiliations:** 1Division of Thoracic Surgery, Department of Surgery, School of Medicine, College of Medicine, Taipei Medical University, Taipei 110, Taiwan; ktkuo@tmu.edu.tw; 2Division of Thoracic Surgery, Department of Surgery, Taipei Medical University—Shuang Ho Hospital, New Taipei City 235, Taiwan; 3Taipei Heart Institute, Taipei Medical University, Taipei 110, Taiwan; chlin99025@tmu.edu.tw; 4Division of Cardiovascular Surgery, Department of Surgery, Shuang Ho Hospital, Taipei Medical University, New Taipei City 235, Taiwan; 5Division of Cardiovascular Surgery, Department of Surgery, School of Medicine, College of Medicine, Taipei Medical University, Taipei 110, Taiwan; 6Department of Dermatology, Taipei Tzu Chi Hospital, Buddhist Tzu Chi Medical Foundation, New Taipei City 231, Taiwan; 10205@s.tmu.edu.tw; 7School of Medicine, Buddhist Tzu Chi University, Hualien 970, Taiwan; 8Division of Urology, Department of Surgery, Faculty of Medicine, Universitas Gadjah Mada/Dr. Sardjito Hospital, Yogyakarta 55281, Indonesia; narpatiwp@gmail.com; 9Department of Medical Research & Education, Taipei Medical University—Shuang Ho Hospital, New Taipei City 235, Taiwan; vijeshp2@gmail.com (V.K.Y.); impossiblewasnothing@hotmail.com (I.-H.F.); 10Department of Medical Laboratory Science and Biotechnology, Yuanpei University of Medical Technology, Hsinchu 300, Taiwan; 11Department of Pathology, Taipei Medical University—Shuang Ho Hospital, New Taipei City 235, Taiwan; whlpath97616@s.tmu.edu.tw; 12Department of Medicine, MacKay Medical College, New Taipei City 252, Taiwan; 13Division of Thoracic Surgery, Department of Surgery, MacKay Memorial Hospital, Taipei 104, Taiwan

**Keywords:** non-small-cell lung cancer, cancer stem cells, human epidermal growth factor receptor 2, NRF2/HO-1/HER2 axis, HNMT/HER2’s role

## Abstract

Background: The treatment of non-small-cell lung cancer (NSCLC) involves platinum-based chemotherapy. It is typically accompanied by chemoresistance resulting from antioxidant properties conferred by cancer stem cells (CSCs). Human epidermal growth factor receptor 2 (HER2) enhances CSCs and antioxidant properties in cancers, including NSCLC. Methods: Here, we elucidated the role of histamine N-methyltransferase (HNMT), a histamine metabolism enzyme significantly upregulated in NSCLC and coexpressed with HER2. HNMT expression in lung cancer tissues was determined using quantitative reverse transcription PCR (RT-qPCR). A publicly available dataset was used to determine HNMT’s potential as an NSCLC target molecule. Immunohistochemistry and coimmunoprecipitation were used to determine HNMT–HER2 correlations and interactions, respectively. HNMT shRNA and overexpression plasmids were used to explore HNMT functions in vitro and in vivo. We also examined miRNAs that may target HNMT and investigated HNMT/HER2’s role on NSCLC cells’ antioxidant properties. Finally, how HNMT loss affects NSCLC cells’ sensitivity to cisplatin was investigated. Results: HNMT was significantly upregulated in human NSCLC tissues, conferred a worse prognosis, and was coexpressed with HER2. HNMT depletion and overexpression respectively decreased and increased cell proliferation, colony formation, tumorsphere formation, and CSCs marker expression. Coimmunoprecipitation analysis indicated that HNMT directly interacts with HER2. TARGETSCAN analysis revealed that HNMT is a miR-223 and miR-3065-5p target. TBHp treatment increased HER2 expression, whereas shHNMT disrupted the Nuclear factor erythroid 2-related factor 2 (Nrf2)/ hemeoxygenase-1 (HO-1)/HER2 axis and increased reactive oxygen species accumulation in NSCLC cells. Finally, shHNMT sensitized H441 cells to cisplatin treatment in vitro and in vivo. Conclusions: Therefore, HNMT upregulation in NSCLC cells may upregulate HER2 expression, increasing tumorigenicity and chemoresistance through CSCs maintenance and antioxidant properties. This newly discovered regulatory axis may aid in retarding NSCLC progression and chemoresistance.

## 1. Introduction

Lung cancer is one of the most common causes of cancer-associated mortality worldwide, accounting for almost 1.76 million death cases per year. Non-small-cell lung cancer (NSCLC) is linked with approximately 80% of all lung cancer cases [1]. Radiotherapy alone or with chemotherapy and adjuvant durvalumab are the most common treatments for patients with locally advanced NSCLC. Platinum-based chemotherapy remains the standard first-line defense treatment for metastatic NSCLC but is frequently accompanied by chemoresistance. Many other chemotherapeutics drugs have been tried as a second-line treatment but the pan-drug resistance acquired by most NSCLC results in treatment failure and uninhibited disease progression. Therefore, developing predictive biomarkers to identify patients and targeted therapies to treat patients likely to benefit from these therapies is critical [2].

Tumor metastasis and chemoresistance have been investigated separately in the past, but these are frequently observed together clinically and are linked biologically. In a study on breast cancer, the interaction between the cancer cells and the host microenvironment was characterized, and the connecting chemotherapy failure with metastatic relapse was observed [3]. Nevertheless, the key molecular mechanisms involved in the association between metastasis and chemoresistance—that might differ in cancer types and clinical settings—remain unknown. Cancer stem cells (CSCs) contribute to tumor relapse, cancer cell propagation, and chemo–radioresistance [4]. They also confer protective antioxidant properties to cancers, leading to chemoresistance [5].

Reactive oxygen species (ROS), including different species of secondary messengers, such as hydrogen peroxide (H_2_O_2_), superoxide anion (O_2_^−^), hypochlorous acid (HOCl), singlet oxygen (^1^O_2_), and hydroxyl radical (OH), are involved in cell signaling for various biological processes in both norms. Weal and cancer cells [6]. ROS regulate critical receptor tyrosine kinase signaling targets, including human epidermal growth factor receptor 2 (HER2), which also directly interacts with NRF2 to regulate DNA binding activity. Furthermore, HER2 downstream effectors include PI3K, and mitogen-activated protein kinase signaling promotes the Nrf2 binding to DNA and results in the transcriptional activation of target genes [7,8].

Histamine, through its interactions with H1–4 receptor subtypes, stimulates various signaling pathways. All these receptors are members of the heptahelical G protein-coupled receptor (GPCR) family [9,10,11]. High histamine-N-methyltransferase (HNMT) expression has been noted in patients with ductal breast cancer compared with healthy controls [12]. Another study reported that a histamine H3 receptor (H3R) antagonist inhibited proliferation and triggered caspase-dependent apoptosis in both estrogen receptor-positive and -negative breast cancer cells [13]. However, the exact mechanism underlying this overexpression and its effect in NSCLC remains unexplored.

In this study, big data, clinical patients, and in vitro and in vivo approaches were used to investigate the role of HNMT in NSCLC. We evaluated HNMT expression in NSCLC and its interaction or coexpression with HER2. Our findings provide the first evidence that the CSCs population in NSCLC has a lower miR3065/223 expression, resulting in a higher expression of their target gene, HNMT. This, in turn, results in a feedback loop mechanism in which HNMT upregulation affects HER2 expression and subsequently increases chemoresistance in NSCLC through tumorsphere formation and the antioxidant response system.

## 2. Results

### 2.1. HNMT Upregulation in NSCLC Tissues Is Related to Worsened Prognosis and Significantly Coexpressed with HER2

To assess the role of HNMT in the progression and development of human NSCLC, we first examined HNMT mRNA expression by using a bioinformatics approach. HNMT mRNA levels increased significantly in patients with NSCLC compared with paired normal adjacent tissue (Figure 1A). We then extrapolated our previous finding to evaluate its prognostic significance by using the Prognoscan database [14]. We observed that patients with higher HNMT expression (*n* = 74) had worse prognoses than did patients with lower HNMT expression (*n* = 37; Figure 1B). HER2 mutation was noted to lead to the worst prognosis in NSCLC. In addition, other studies have indicated a potential interaction between HNMT and HER2 [15]. Therefore, we investigated the interaction between them in NSCLC. We used the R2 database [16] and three datasets—namely Peitsch (*n* = 121), EXPO (*n* = 150), and Bild (*n* = 114)—and detected that HNMT and HER2 were significantly coexpressed (Figure 1C–E). We verified HER2 coexpression with HNMT in tissues obtained from patients with NSCLC (Figure 1F). Moreover, HER2 was not more significantly expressed in NSCLC tumor tissues, whereas HNMT was strongly expressed in the tumor tissue compared with the normal adjacent tissue (Figure 1G). Notably, according to our bioinformatics findings, both HER2 and HNMT were significantly coexpressed in the clinical samples (Figure 1H). Furthermore, we validated the standard to compare the clinicopathologic parameters, as shown in Table 1 (below), we found that tumor differentiation, tumor size, lymph node metastasis (LMN) status, and pathological clinical-stage remained significantly different between HNMT high and low expression group. Additionally, the cell type was also a significant factor, with HNMT high expression less frequent in NSCLC.

Subsequent univariate (UA) and multivariate analysis (MA) was used to evaluate the significance of the expression of HNMT and Her2, age, gender, tumor size, LNM, clinical-stage, subtypes on NSCLC-specific survival. Overexpression of HNMT (hazard ratio 0.294, 95% confidence interval: 0.108 to 0.806, *p* < 0.017 for UA and hazard ratio 0.152, 95% confidence interval: 0.047 to 0.495; *p* < 0.002 for MA, Table 2) could be considered as independent prognostic factors in NSCLC patients. Taken together, these results imply that HNMT is associated with HER2 expression and is a potential therapeutic target in NSCLC.

### 2.2. HNMT Interacts with HER2 and Affects NSCLC Cell Line Development

We next examined HNMT expression in six NSCLC cell lines (BEAS-2B, CL1-0, CL1-5, H838, A549, and H441) through Western blotting. The results revealed that HNMT protein levels were significantly upregulated in A549 and H441 cells but were mostly underexpressed in other NSCLC cell lines. Notably, HNMT expression was accompanied by HER2 and HER3 expression. The cells with higher HNMT expression also had higher vimentin but lower N-cadherin expression, implying HNMT’s role in EMT in NSCLC (Figure 2A). Therefore, we employed H441 and CL1-0 cells for loss-of-function and gain-of-function tests to elucidate the role of HNMT expression in NSCLC. Figure 2B,C depicts the effectiveness of the clones used for HNMT knockdown or overexpression in H441 and CL1-0 cells, respectively. Because we detected HER2 coexpression with HNMT in NSCLC cell lines, we investigated the association between the two. Reciprocal immunoprecipitation with HNMT or HER2 antibodies indicated a powerful interaction between endogenous HNMT and HER2 in the H441 cells (Figure 2D). HNMT also affected NSCLC cell proliferation (Figure 2E,F); its knockdown was associated with reduced cell proliferation, and its overexpression was associated with an increased cell proliferation rate. Raw data of western blot with full-size blot shown in Appendix A. The results of these experiments further confirm the interaction between HNMT and HER2 and their role in NSCLC cell development.

### 2.3. HNMT Promotes CSC Development in NSCLC Cell Lines

To evaluate whether HNMT also played a role in the origin of cancer stemness in NSCLC, we first assessed the colony formation ability of H441 with HNMT knockdown and of CL1-0 with HNMT overexpression. We observed that HNMT knockdown negatively influenced H441 colony formation. Its overexpression, however, significantly increased the colony formation ability of CL1-0 cells (Figure 3A). Similarly, HNMT knockdown and overexpression resulted in reduced and increased, respectively, tumorsphere formation ability in H441 and CL1-0 cells lines (Figure 3B). HNMT manipulation also affected stemness marker expression, which was evaluated through Western blotting. In H441 cells, shHNMT led to significant downregulation of KLF4, NANOG, OCT4, and CD133 compared with shScramble. HNMT overexpression, in turn, increased KLF4, NANOG, OCT4, and CD133 expression in CL1-0 cells (Figure 3C), raw data of western blot with full-size blot illustrated in Appendix A.

### 2.4. HNMT Is the Gene Target of miR-223 and miR-3065-5p

We next examined whether miRNA may target HNMT using TARGETSCAN (version 6.2) and observed that HNMT contained two complementary 7mer target sites for miR-223 and one site for miR-3065-5p (Figure 4A). We evaluated miR-223 and miR-3065-5p expression in NSCLC tumorspheres and found both expressions were significantly decreased in NSCLC tumorspheres compared with that in control adherent cells (Figure 4B). Furthermore, the H441 cells were transfected with the mimics of miR-223, miR-3065-5p, or both. The cells were probed for HNMT and HER2 expression analysis through immunofluorescence. As expected, a single miRNA mimic for either miR-223 or miR-3065-5p decreased HNMT expression and combining both the mimics further decreased HNMT expression (Figure 4C). Next, we transfected H441 cells with the wild-type (WT) or mutated (mut) HNMT 3′UTR-directed luciferase reporter, and we then cotransfected the cells with miR-223, miR-3065-5p mimic, or both. Through this experiment, we demonstrated that WT HNMT 3′UTR luciferase activity was suppressed with miRNA mimics cotransfection, but mut HNMT 3′UTR did not substantially affect miRNA mimics. This result further affirmed that HNMT is the target gene of miR-223 or miR-3065-5p (Figure 4D). Western blotting also yielded similar results for HNMT, and HER2 expression significantly decreased with either miR-223 or miR-3065-5p mimic transfection, with further reduction with transfection of both mimics (Figure 4E), Raw data of western blot with full-size blot displayed in Appendix A. MiR-223 and miR-3065-5p mimic transfection also negatively influenced H441 colony formation, which further reduced shHNMT cotransfection (Figure 4F).

### 2.5. HNMT Regulates the Antioxidative Stress Pathway in NSCLC Cells

ROS accumulation has been employed as a cancer-killing mechanism in chemotherapeutic development. HER2 expression has been shown to confer protection against ROS damage in cancer cells; thus, targeting HER2 activity in cancer cells may increase their susceptibility to chemotherapy. We first observed increased HER2 promoter activity after treatment with various dosages of TBHp (50, 100, or 200 µM). Next, we transfected H441 cells with shHNMT and treated them with 200 μM TBHp. As expected, HER2 promoter activity significantly decreased in cells with HNMT knockdown (Figure 5A). Next, to confirm that HER2 and HER3 expression increases under oxidative stress conditions, we treated H441 cells with 200 µM TBHp and found that TBHp increased HER2 and HER3 expression compared with untreated cells (Figure 5B). We then evaluated intracellular ROS levels by using DCFDA, a fluorescence probe, and found that HNMT knockdown significantly increased ROS generation (Figure 5C). HNMT knockdown also significantly disrupted the activity of HER2 via NRF2/HO-1 signaling as analyzed through Western blotting (Figure 5D), raw data of western full-size blot shown in Appendix A.

### 2.6. HNMT Inhibition Sensitizes NSCLC to Cisplatin Chemotherapy Both at In Vivo and In Vitro 

Our in vitro assays indicated that HNMT/HER2 signaling plays a vital role in cancer stemness and antioxidant pathways. Thus, we hypothesized that targeting HNMT expression sensitizes NSCLC cells to conservative chemotherapy. We first compared the viability of H441 cells transfected with shScramble and shHNMT and treated with an increasing cisplatin dosage. HNMT knockdown sensitized H441 cells to cisplatin treatment (Figure 6A). We then evaluated the apoptosis of cells by using Annexin V/PI staining and noted that H441 cells transfected with shHNMT had a higher cell apoptosis rate compared with those transfected with shScramble (Figure 6B). Subsequently, the H441 cells transfected with shHNMT and treated with 4 µM cisplatin had greatly reduced colony formation ability compared with either cisplatin-only or shHNMT-only cells (Figure 6C). Next, we extrapolated ours in vitro results to the conducted in vivo trials. After all, mice were sacrificed on day 22 it was discovered that shHNMT sensitized the tumor to cisplatin treatment (Figure 6D). However, we identified no significant weight difference between the groups, implying that shHNMT knockdown was, at least partially, safe (Figure 6E). We then evaluated the tumor tissue through the TUNEL assay. We noted that tumors with HNMT knockdown and treated with cisplatin had an increased level of TUNEL staining, suggesting an increased apoptotic rate (Figure 6F). Combined, these data suggest that shHNMT has a vital role to play in conferring chemoresistance to NSCLC cells.

## 3. Discussion

Here, we discovered that the HNMT was substantially upregulated in NSCLC tissues, in accordance with our findings obtained through the use of the Prognoscan database data. Furthermore, patients with high HNMT expression had the worst prognosis. We examined three datasets using the R2 platform, and we noted that HNMT was significantly associated with HER2 expression. We confirmed this finding in the clinical NSCLC samples, which exhibited higher HNMT and HER2 expression in the tumor tissue than in the nontumor tissue. To further elucidate the role of HNMT in NSCLC tumorigenesis, we employed NSCLC cell lines for loss-of- and gain-of-function studies. Our co-immunoprecipitation data indicated that HNMT interacted with HER2, and its manipulation affected NSCLC cell growth rates. Identifying the role of HNMT in the CSCs subpopulation in NSCLC, we assessed the colony formation ability of NSCLC cell lines with HNMT knockdown or overexpression. HNMT loss or gain significantly reduced or increased NSCLC colony formation, respectively. Consistently, the cell lines’ tumorsphere formation was similarly affected by HNMT loss/gain. CSCs markers, such as KLF4, NANOG, OCT4, and CD133, were increased and decreased following HNMT knockdown and overexpression, respectively. Next, we explored the regulation of HNMT expression in NSCLC by using TARGETSCAN data to determine which miRNA may target this molecule. Results indicate that the 3′UTR region of HNMT has been a complementary site of miR-3065-5p and miR-223. Notably, miR-3065 and miR-223 had relatively low expression in the NSCLC tumorsphere population. Using inhibitors and mimics of both miR-3065 and miR-223, we observed that HNMT was the target of both miRNAs.

HER2 endows cancer cells with robust antioxidant properties [8,17]. Given the HNMT–HER2 interaction, we assessed the effect of TBHp-induced oxidative damage on NSCLC cells with HNMT knockdown. Our luciferase reporter assay confirmed that various TBHp dosages increased HER2 and HER3 signal intensity, but HNMT knockdown significantly reduced their intensity. We also detected a higher rate of cells with DCFDA staining, implying higher ROS accumulation, in H441 cells with HNMT knockdown. Finally, our Western blotting results indicated that shHNMT reduced the HER2/NRF2/HO-1 signaling pathway activity. CSCs and cancer cells’ antioxidative properties contribute to chemoresistance. Having proven that HNMT plays a significant role in CSCs maintenance and oxidative stress response, we subsequently treated WT and shHNMT H441 cells with varied dosages of cisplatin and evaluated their response. HNMT knockdown increased H441 cells’ sensitivity to cisplatin treatment, as evidenced by their lower viability and higher apoptosis rate. In addition, cisplatin significantly reduced shHNMT H441 colony formation compared with shScramble H441. We then extrapolated these in vitro findings by transfecting shScramble and shHNMT H441 cells into NOD/SCID mice and evaluating their response to cisplatin treatment. Mice bearing shHNMT H441 cells were more sensitive to cisplatin treatment as indicated by a delayed growth rate and increased TUNEL staining intensity compared with the shScramble group.

Histamine has numerous biological activities, including roles in cell proliferation and differentiation, gastrointestinal function regulation, and immune response modulation [18]. It is a low-molecular-weight amine generated solely by L-HDC and is found in various cells in the body, including the gastric mucosa and parietal and mast cells [19]. Histamine acts by interacting with its GPCRs—namely H1HR, H2HR, H3HR, and H4HR. When these receptors are activated or inhibited, downstream signaling pathways are triggered, eliciting immune-modulatory and proinflammatory cell responses [20]. After histamine is produced by HDC, it is rapidly stored or degraded by HNMT and monoamine oxidase B [21,22].

The role of HNMT in cancer remains unclear. A microarray-based study on esophageal squamous cell carcinoma determined that HNMT is one of the crucial players in controlling signal transduction networks [23]. In pediatric acute lymphoblastic leukaemia, 75 key genes were identified, all of which were considerably enriched in 25 GO functions and the chronic myeloid leukaemia pathway. Subsequently, 27 disease risk subpathways were identified with HNMT as a key gene enriched in these subpathways [24]. B[a]P is a well-known polycyclic aromatic hydrocarbon and a common pollutant in the atmosphere that can cause cancer in both animals and humans. A study conducted in HepG_2_ cells revealed that HNMT gene expression significantly increased with B[a]P treatment; however, its precise role in carcinogenesis remains unclear [25].

In this study, we elucidated the role of HNMT in cancer, specifically in NSCLC carcinogenesis. Histamine metabolism may play a major role in NSCLC carcinogenesis [26,27]. The direct interaction of HNMT with HER2 at least partially explains the carcinogenic role of HNMT. HER2 overexpression has been observed in patients with many solid cancers, including NSCLC. Although the exact mechanism underlying HER2–HNMT interaction remains unclear, HNMT may be associated with HER2 homodimerization, as indicated by Yoshioka et al. [28]. The authors also demonstrated that SET and MYND domain-containing protein 3 (SMYD3), a protein lysine methyltransferase, can affect HER2 homodimerization and the activation of its downstream pathways by inducing the trimethylation of lysine 175 residues in HER2 [28].

HER2 is a well-known prognostic and predictive factor in breast cancer; however, its function in lung cancer warrants further clarification [15]. HER2 mutations can be found in a small percentage of patients with lung cancer. These shifts can be interpreted as oncogenic drivers as well as a mechanism of acquired resistance following targeted therapy. Similar changes have been observed in other cancers, such as breast and gastric cancer, and they have been linked to poor prognosis and short overall survival [29,30,31,32]. In addition, HER2-altered stage IV NSCLC leads to a relatively short overall survival, presumably due to intrinsic resistance to chemotherapy [33]. Consistent with our findings, HER2 expression has also been implicated in CSCs’ development. According to previous data, HER2 may enhance carcinogenesis, invasion, and metastasis in HER2-positive breast cancers, at least in part, by sustaining and increasing CSCs [34,35]. Honkanen et al. found that, in patients with NSCLC, HER2 modulates the CSLC phenotype in ALK-translocated lung cancers, and this modulation is primarily orchestrated by HER2/HER3 heterodimers [36]. HER2 was also implicated in CSCs-induced chemoresistance [37]. Wang et al. [37] investigated the effect of HER2 on the induction of CSCs and the drug susceptibility of ovarian cancer cell lines. HER2 expression was correlated with tumorsphere formation efficiency, and NF-κβ was responsible for the mediation of HER2-induced CSCs. Furthermore, HER2 inhibition significantly increased the sensitivity of ovarian cancer cells [37]. In addition, CSCs may confer antioxidant properties that may interfere with chemotherapeutic responses in cancers [38]. As mentioned above, we observed that HNMT knockdown disrupts the HER2/NRF2/HO-1 signaling axis. This is in accordance with previous studies that demonstrated HER2–NRF2 interaction induces oxaliplatin resistance in colon and breast cancer cells [5,8].

CSCs can undergo significant changes that may alter their biological signature, including their miRNA expression. MiRNAs are often found to be dysregulated, and they may strongly affect tumorigenicity. In our study, miR-223 and miR-3065 were significantly downregulated in H441 CSCs populations. This finding validated our preliminary findings because we also demonstrated through TARGETSCAN database mining that these two miRNAs were highly correlated with HNMT. Therefore, these data—at least partially—explained why HNMT is more highly expressed in sphere-enriched H441 cells. The role of miRNAs in HNMT regulation remains underexplored. A study demonstrated that miR-223 might indirectly upregulate HNMT expression in atopic dermatitis pathogenesis [39]. By contrast, we observed that miR-223 inhibited HNMT expression. A recent review elaborated the complexities of this miRNA: it can act as either an oncomir or oncosuppressor [40]. Unlike miR-223, data concerning the role of miR-3065 in cancer have been relatively limited; however, miR-3065 was recently found to be a potential predictor of disease severity in ovarian cancer, along with its target ADH7 [41]. MiR-3065 is also associated with renal cancer cell tumorigenicity [42]. Our study adds to existing evidence for a miR-3065′s role as an oncosuppressor that interacts with HNMT in lung cancer.

## 4. Methods

### 4.1. Clinical Samples

The microarray gene expression datasets of the lung cancer patients were analyzed by using the online Prognoscan database [Duke (*n* = 111)] to generate the Kaplan–Meier survival curve [14]. Gene correlation analysis of HNMT and HER2 was analyzed using R2 Genomic Analysis and Visualization Platform (https://hgserver1.amc.nl/cgi-bin/r2/main.cgi; accessed on 21 November 2021) by using the Peitsch (*n* = 121), EXPO (*n* = 150), and Bild (*n* = 114) lung cancer datasets. Furthermore, the patients’ clinical samples were also collected from MacKay Memorial Hospital, Taipei City, Taiwan. All patients provided written consent for their tissue to be used for scientific research. The study of patients’ samples was approved by the MacKay Memorial Hospital (Approval no.: IRB:20MMHIS500e) and complied with the recommendations of the Declaration of Helsinki for Biomedical Research. In all patients, lung cancer tissues and normal tissues >3 cm away from the cancer were obtained. Tissue arrays of NSCLC samples were subjected to immunohistological analysis after incubating with primary antibodies against the HNMT (1:100 dilution, SC-81159; Santa Cruz Biotechnology, USA) and HER2 (1:100 dilution, SC-81159; Santa Cruz Biotechnology, USA) at 4 °C overnight. HRP and DAB staining with hematoxylin counterstaining were performed as per the standard immunohistochemistry protocol were followed by imaging and estimation of the expression of the protein.

### 4.2. Cells and Culture Medium

The lung cancer cell lines, such as BEAS-2B, CL1-0, CL1-5, H838, A549, and H441, were purchased from the American Type-Culture-Collection (ATCC, Manassas, VA, USA). The cells were cultured in Dulbecco’s modified Eagle’s medium (#12491023; GIBCO, Life Technologies, Carlsbad, CA, USA) supplemented with 10% fetal bovine serum (GIBCO, Life Technologies, Carlsbad, CA, USA), penicillin (100 IU/mL), and streptomycin (100 g/mL; #15140122, GIBCO, Life Technologies, Carlsbad, CA, USA) grown in the humidified, 5% CO2 incubator at 37 °C.

### 4.3. Cell Stable Transfection 

The information of the HNMT expression plasmid (OriGene RC204676) was used to design polymerase chain reaction (PCR) primers. Oligonucleotides Hind III-HNMT -F (5′-AAT TAA GCT TAT GGC ATC TTC CAT GAG GAG-3′) and MluI-HNMT-R (5′-AAT TAC GCG TTG CCT CAA TCT CTA TG-3′) were designed for PCR amplification of HNMT gene sequences. The segment of HNMT was borne on an empty plasmid (pCMV-MCS-N1 Empty vector control plasmid DNA, GenBank accession U55762). Cell transfection was performed using Lipofectamine 2000 (Invitrogen), following the manufacturer’s protocols. 

Eight micrograms of empty plasmid (pCMV6-Entry vector control plasmid DNA, OriGene accession PS100001) or HNMT expression plasmid (pCMV6-HNMT, OriGene RC204676, HNMT#1; pCMV-HNMT-N1, GenBank accession U55762, HNMT#2) were used. The DNA-lipofectamine reagent complexes stay at room temperature for 30 min. The mixture was added to the well and mixed gently by rocking the plate back and forth. Reagent complexes did not have to be removed following transfection. The cells were incubated at 37 ℃ in a CO_2_ incubator for 48 h and assayed for transgene expression.

### 4.4. Co-Immunoprecipitation (Co-IP) 

Co-immunoprecipitation was used to detect HNMT–HER2 interaction *in vitro*. The standard Co-IP protocol was the same as that described for IP. Non-denaturing lysis buffer (20 mM Tris HCl pH 8, 137 mM NaCl, 1% Nonidet P-40, 2 mM EDTA) was stored at 4 °C, and immediately before use, protease inhibitors were added. The cell culture was placed in a dish on ice, and the cells were washed with ice-cold PBS. Then ice-cold lysis buffer was added. After that, cells were completely lysed under non-denaturing conditions, and proteins that bound together were kept. Irrelevant, non-binding proteins, antigens, and any proteins that were bound were eluted by a series of washes. Then, the bound proteins which eluted were analyzed by SDS-PAGE/immunoblotting.

### 4.5. Western Blotting

Lung cancer cells were extracted and lysed after trypsinization. After the total proteins lysates were extracted and the sample was prepared, it was separated using the SDS-PAGE using Mini-Protean III system (Bio-Rad, Taiwan) and transferred onto PVDF membranes using Trans-Blot Turbo Transfer System (Bio-Rad, Taiwan). Membranes were incubated overnight at 4 °C in the primary antibodies shown in Appendix A. Secondary antibodies were purchased from Santa Cruz Biotechnology (Santa Cruz, CA, USA), and an ECL detection kit was used for the detection of the protein of interest. Images were captured and analyzed using the UVP BioDoc-It system (Upland, CA, USA).

### 4.6. Total RNA Isolation and Qunatitiave Reverse-Transcription Polymerase Chain Reaction (qRT-PCR) 

The total RNA was isolated and purified using TRIzol-based protocol (Invitrogen, ThermoFisher Scientific, Waltham, MA, USA) according to the protocol provided by the manufacturer. Two microgram of total RNA was reverse transcribed using QIAGEN OneStep RT-PCR Kit (QIAGEN, Taiwan), and the PCR reaction was performed using a Rotor-Gene SYBR Green PCR Kit (400, QIAGEN, Taiwan). HNMT mRNA expression was detected in lung cancer and normal tissues. The primer sequences used were as follows: HNMT amplification (452 bp), 5′-TACGTCCAAGGTCGGGCAGGAAGA-3′; upstream, 5′-CACTGATAGGCAGTTCTC; downstream, 5′-GGTTCTCAGTTGGTGCTTC. Glyceraldehyde-3-phosphate dehydrogenase (GAPDH) was used as an internal reference to detect HNMT mRNA expression levels. The relative expression level of HNMT mRNA was calculated using 2−∆∆Cq formulae. 

### 4.7. Clonogenic Assay

For the assessment of the sensitivity of cancer cells towards any treatment, the “Clonogenic Assay” is a gold standard. A total of 2.7 × 10^4^ cells per well were seeded in a 6-well plate and incubated at 37 °C for 2 days. Further, the cells were cultured for an additional 24 h in media and incubated at 37 °C for 2 days in 5% CO_2_ treated. The cells were then subcultured and seeded at 350 cells per well into new 6-well plates and kept for incubation for 10 days at 37 °C in a humidified incubator with 5% CO_2_. The cells were fixed and dried after being set and stained with 0.1% crystal violet. The experiments were conducted in triplicate.

### 4.8. Sulforhodamine B Assay

Cellular viability was determined using the sulforhodamine B (SRB) assay as per the protocol suggested by our lab protocol. Briefly, lung cancer cells were seeded in 96-well plates (3.5 × 10^5^ cells /well), followed by incubation at 37 °C, in a humidified 5% CO_2_ incubator. The cells were fixed with the gentle addition of 50 mL of cold 10% *w*/*v* tricarboxylic acid (TCA) and incubated at 4 °C for 60 min. Next, 50 μL of 0.4% *w*/*v* SBR solution in 1% CH3COOH was added to each well, followed by incubation at room temperature for 20 min. Unbound dye was recovered after staining, and residual dye was removed by thoroughly washing well plates with 1% CH3COOH and air drying. The bound stain was dissolved in 10 mM Trizma base, and the absorbance was measured at 515 nm on an ELISA plate reader (690 nm reference wavelength).

### 4.9. Immunofluorescence Assay

H441 cells were cultured on glass coverslips before being transfected, as mentioned previously. After incubation, cells were fixed for 15 min at 4 °C with 4% formaldehyde, permeabilized for 5 min with 0.01% Triton X-100 and blocked for 30 min at room temperature with 1% bovine serum albumin. The cells were then incubated for 24 h at 4 °C with primary antibodies against GSK-3 (#12456, 1:100, Cell Signaling Technology, Danvers, MA, USA) and β-catenin (#8814, 1:100, Cell Signaling Technology). The cells were stained with an isotype-specific secondary antibody (Alexa Fluor 594-AffiniPure donkey antirabbit IgG; Jackson ImmunoResearch, West Grove, PA, USA) for 1 h the following day.

### 4.10. ROS Production Measurement

For ROS production evaluation, cancer cells were plated in 96-well plates at 20,000 cells per well in a final volume of 80 μL of the medium. Next, 10 μL of 50 μM DCFDA was added to each well, and the cells were incubated for 30 min. On a microplate fluorometer (Tecan, Seestrasse, Männedorf, Switzerland), fluorescence was measured with an excitation filter set at 488 nm and an emission filter set at 530 nm.

### 4.11. Tumor Xenograft Study

Four-to-six-week-old female NOD/SCID mice (mean weight = 17.4 ± 2.1 g) were purchased from BioLASCO Taiwan (Taipei, Taiwan). The in vivo studies were approved by the Institutional Animal Care and Use Committee (IACUC) of the MacKay Memorial Hospital (Approval no.: MMH-A-S-109-10). The mice (*n* = 10 each) were randomly assigned to the shScramble, shScramble + cisplatin, shHNMT, or shHNMT + cisplatin groups after receiving an injection of 2 × 10^6^ H441 cells with scramble shRNA (shScramble) or HNMT shRNA (shHNMT) in their hind flanks. When the tumors were palpable on day 8, 2 mg/kg cisplatin was administered intraperitoneally every 72 h for 12 days. Tumor sizes were determined with callipers every 3 days on days 6, 9, 12, 15, and 18, and tumor volumes (v) were estimated using the formula length (l) × (width (w))2 × 0.5. The tumor-bearing mice were humanely sacrificed at the end of the trial on day 18, and the tumors were removed, tested, photographed, and weighed again. 

### 4.12. Statistical Analysis

The means and standard deviations (SDs) were used to present all results. For multiple comparisons or repeated measurements, Student’s *t*-test was used. For multiple comparisons or repeated measurements, ANOVA or repeated ANOVA accompanied by Tukey’s post hoc test was used. Statistical significance was defined as *p* < 0.05. GraphPad Prism (version 7; GraphPad Software, San Diego, CA, USA) was used for all statistical analyses.

## 5. Conclusions

Our results revealed that HNMT upregulation in NSCLC cells leads to HER2 upregulation, which in turn increases tumorigenicity and chemoresistance through CSCs maintenance and antioxidant properties. This CSCs may downregulate miR-3065-5p and miR-223 expression, thus reducing the inhibition of their target gene, HNMT, thereby resulting in a feedback loop that may aid in maintaining the CSCs population of NSCLC and conferring chemoresistance (Figure 7). These results provide novel insight into the roles and interactions of HNMT, HER2, and miRNAs in NSCLC pathogenesis and behaviour. Targeting this newly discovered regulatory axis may aid in retarding NSCLC progression and combating chemoresistance.

## Figures and Tables

**Figure 1 ijms-23-01663-f001:**
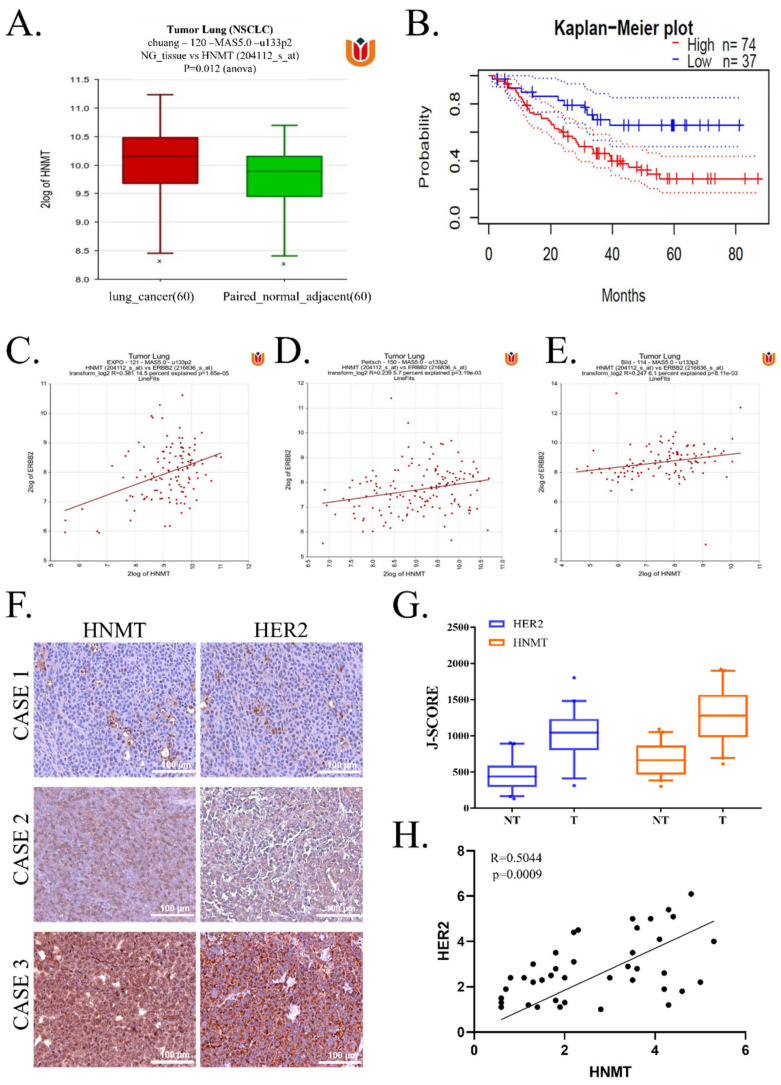
Association of HNMT with worse prognosis in NSCLC and its correlation with HER2 expression. (**A**) HNMT mRNA expression was evaluated using RT-qPCR. Samples were grouped as NSCLC and paired with normal adjacent. (**B**) Kaplan–Meier survival analysis of patients with NSCLC expressing HNMT based on the Prognoscan database (GSE3141). HNMT and HER2 (ERBB2) gene correlation analysis determined using R2: Genomics Analysis and Visualization Platform (http://r2.amc.nl; accessed on 21 November 2021) from the dataset (**C**) Peitsch (*n* = 121), (**D**) EXPO (*n* = 150), and (**E**) Bild (*n* = 114). (**F**) Representative immunohistochemistry staining from three NSCLC cases. (**G**) J-score comparison between nontumor and tumor tissues taken from NSCLC clinical tissue samples. (**H**) HNMT and HER2 protein expression correlation analysis in NSCLC clinical tissue samples. Scale bar: 100 μm.

**Figure 2 ijms-23-01663-f002:**
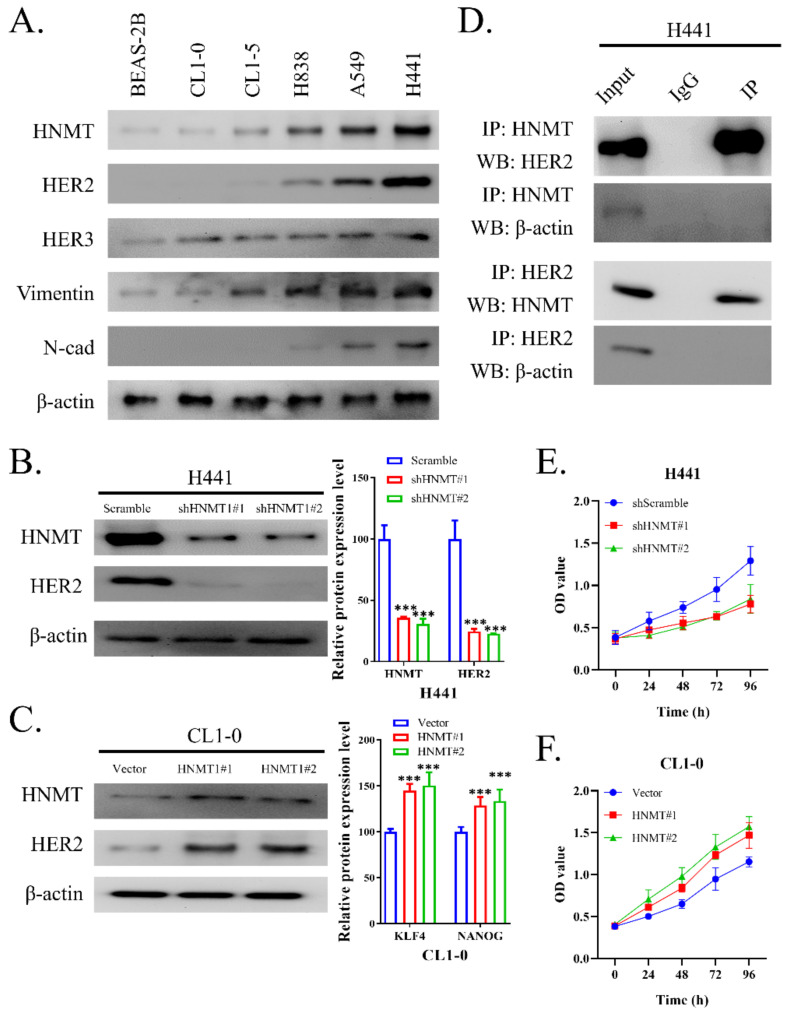
Molecular manipulation effectiveness of HNMT in lung cancer cell lines. (**A**) Western blotting of NSCLC cell lines. Cell lysates from six NSCLC cell lines have been subject to Western blotting. HNMT, HER2, HER3, vimentin, and N-cad were detected using their corresponding antibodies. The experiments were carried out after calibration using GAPDH as an internal control. (**B**) Western blotting for the HNMT and HER2 expression in H441 cells transfected with shScramble, shHNMT#1, or shHNMT#2. (**C**) Western blotting for HNMT and HER2 expression of CL1-0 cells transfected with control vector, HNMT-overexpressing plasmid #1 (pCMV6-HNMT, HNMT#1), or HNMT-overexpressing plasmid #2 (pCMV-HNMT-N1, HNMT#2). Densitometry analysis for the protein expression of both cell lines is provided. (**D**) Coimmunoprecipitation was used to detect HNMT–HER2 interaction and β-actin were used as a housekeeping control. (**E**) Cell proliferation growth curve comparison of H441 cells transfected with shScramble, shHNMT#1, or shHNMT#2, and (**F**) CL1-0 cells transfected with vector control or control vector, HNMT-overexpressing plasmid #1 (HNMT#1), or HNMT-overexpressing plasmid #2 (HNMT#2). **** p* < 0.001.

**Figure 3 ijms-23-01663-f003:**
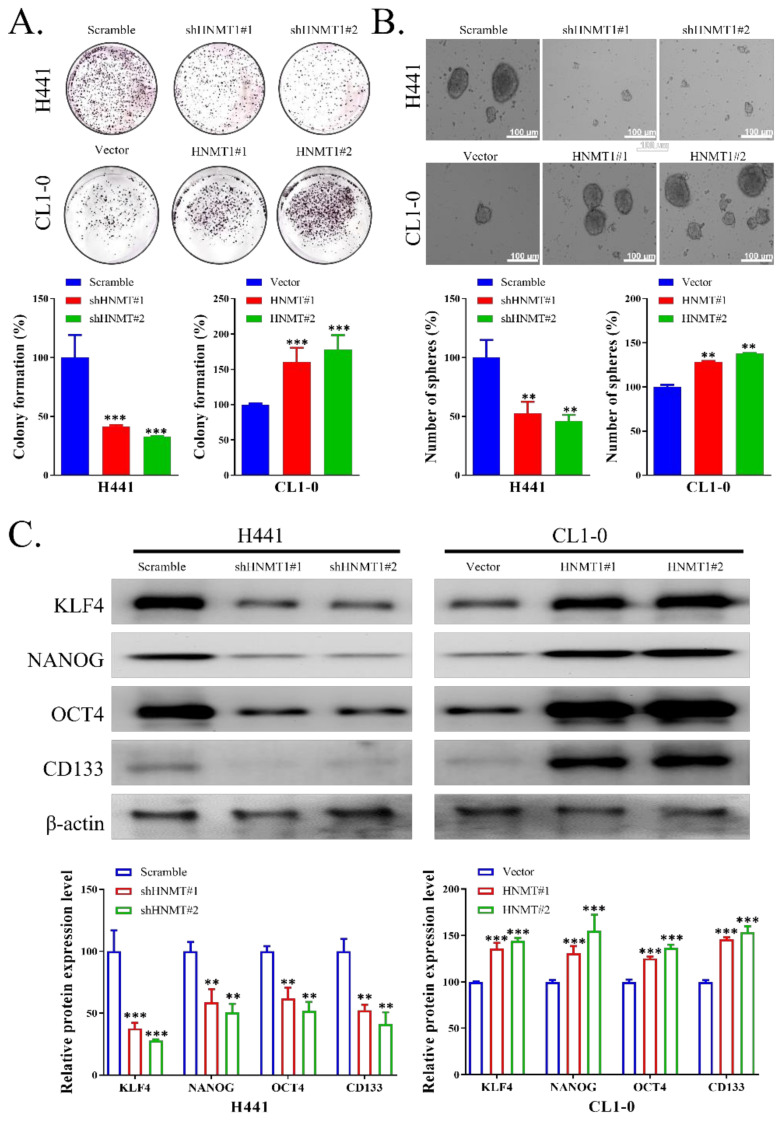
Effects of HNMT on CSCs maintenance in lung cancer cell lines. (**A**) H441 and CL1-0 cells transfected with knockdown and expression vectors were seeded on 6-well plates. Subsequent to 2 weeks, the colonies were stained with crystal violet, and the number of colonies was counted. (**B**) Tumorsphere formation potential of NSCLC cells. Both H441 and CL1-0 cells could form tumorspheres; HNMT knockdown and overexpression significantly reduced and increased their tumorsphere-forming potential, respectively. (**C**) Cell lysates from the H441 and CL1-0 cells with HNMT knockdown and overexpression were subjected to Western blotting, respectively. HNMT, HER2, HER3, vimentin, and N-cad were detected using their respective antibodies. The experiments were conducted following calibration using β-actin as the internal housekeeping control. The experiments were analyzed and normalized across three separate experiments. ** *p* < 0.01, *** *p* < 0.001. Scale bar: 100 μm.

**Figure 4 ijms-23-01663-f004:**
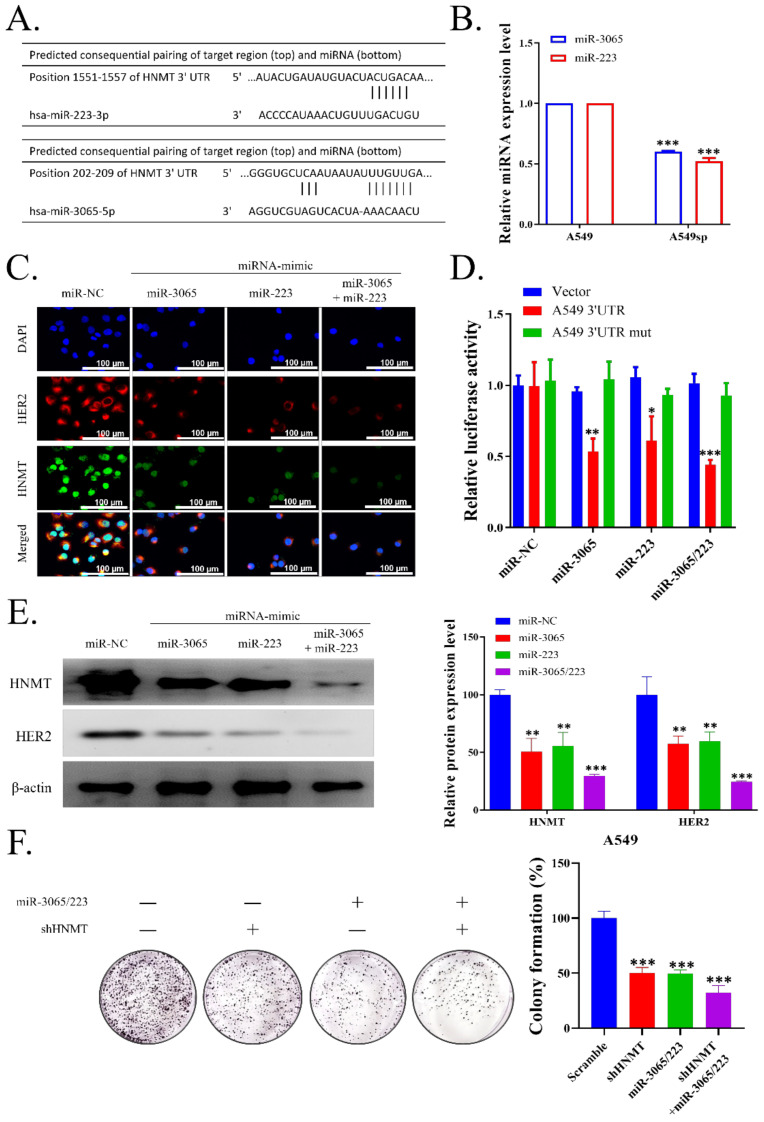
HNMT is one of the key target genes of miR-3065 and miR-223 in lung cancer. (**A**) Complementary sites for miR-3065-5p and miR-223 in the 3′UTR region of HNMT according to TARGETSCAN (version 6.2). (**B**) miR-3065-5p and miR-223 expression in adherent versus sphere H441 cells. RNU 6-2 was used as control. Both qPCR results were normalized to adherent H441 value. (**C**) Immunofluorescence staining of HNMT and HER2 in H441 cells transfected with NC or miR-3065 and/or miR-223 inhibitor demonstrated differential localization. DAPI staining (blue) was used to label DNA in all cells. Scale bar = 100 µm. (**D**) H441 cells were cotransfected with miR-3065, miR-223, or control mimic (miR-NC) and wild-type (WT) or mutated (mut) HNMT 3′UTR-directed luciferase reporter. Luciferase activity was measured using dual-luciferase reporter assays. (**E**) Western blotting of HNMT and HER2 differential protein expression in H441 cells transfected with NC or miR-3065 and/or miR-223 inhibitor. (**F**) H441 cells transfected with NC or miR-3065 and/or miR-223 mimics showing differential colony formation potential. Graph bars are mean ± SDs of three independent experiments. ** p* < 0.05, *** p* < 0.01, **** p* < 0.001. Scale bar: 100 μm.

**Figure 5 ijms-23-01663-f005:**
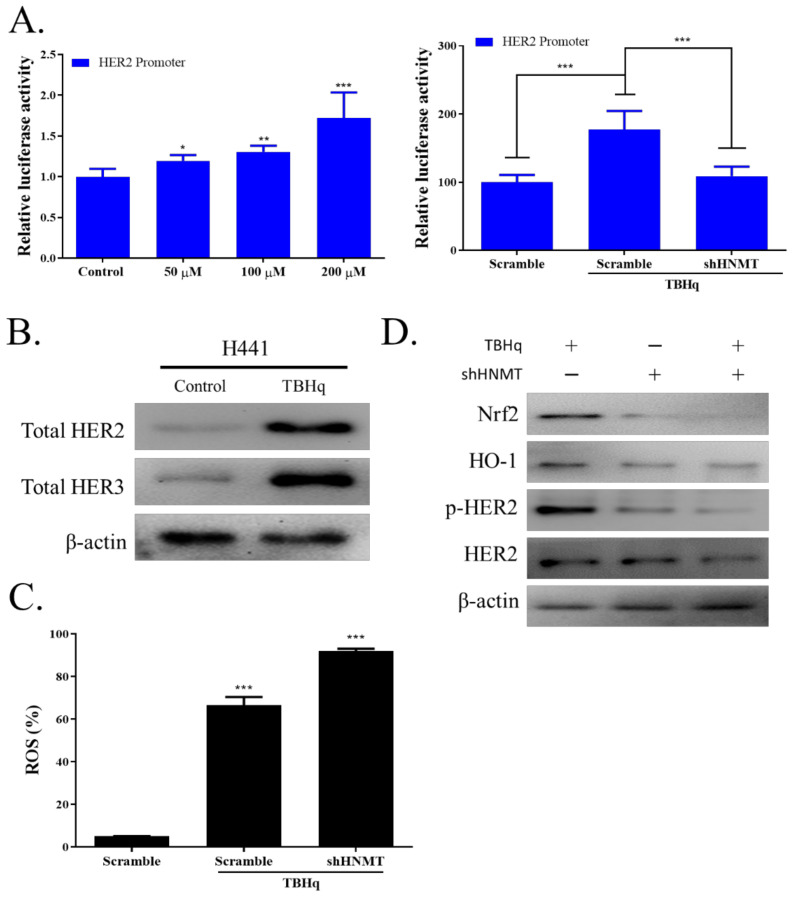
The negative influence of HNMT knockdown on HER2-dependent antioxidant response mechanism in lung cancer cells. (**A**) tBHQ-induced HER2 promoter transcriptional activity induction in a concentration-dependent manner. The H441 cell line was transfected and then treated for 4 h with various concentrations of tBHQ, as indicated. (**B**) Following tBHq treatment, immunoblot analysis showed total protein induction of both HER2 and HER3. Cells were treated with 200 µM tBHQ for 4 h, lysed to extract the protein, and analyzed through Western blotting. (**C**) Flow cytometric analysis of tBHQ-induced ROS formation in H441 cells with or without shHNMT transfection using the ROS-sensitive fluorometric probe DCFDA. Relative ROS production in all the tested cells was normalized to the TBH1 + shHNMT group. (**D**) Immunoblot analysis of cells treated with 200 µM tBHq with or without shHNMT transfection. NRF2, HO-1, pHER2, and HER2 were detected with their respective antibodies. ** p* < 0.05, *** p* < 0.01, **** p* < 0.001.

**Figure 6 ijms-23-01663-f006:**
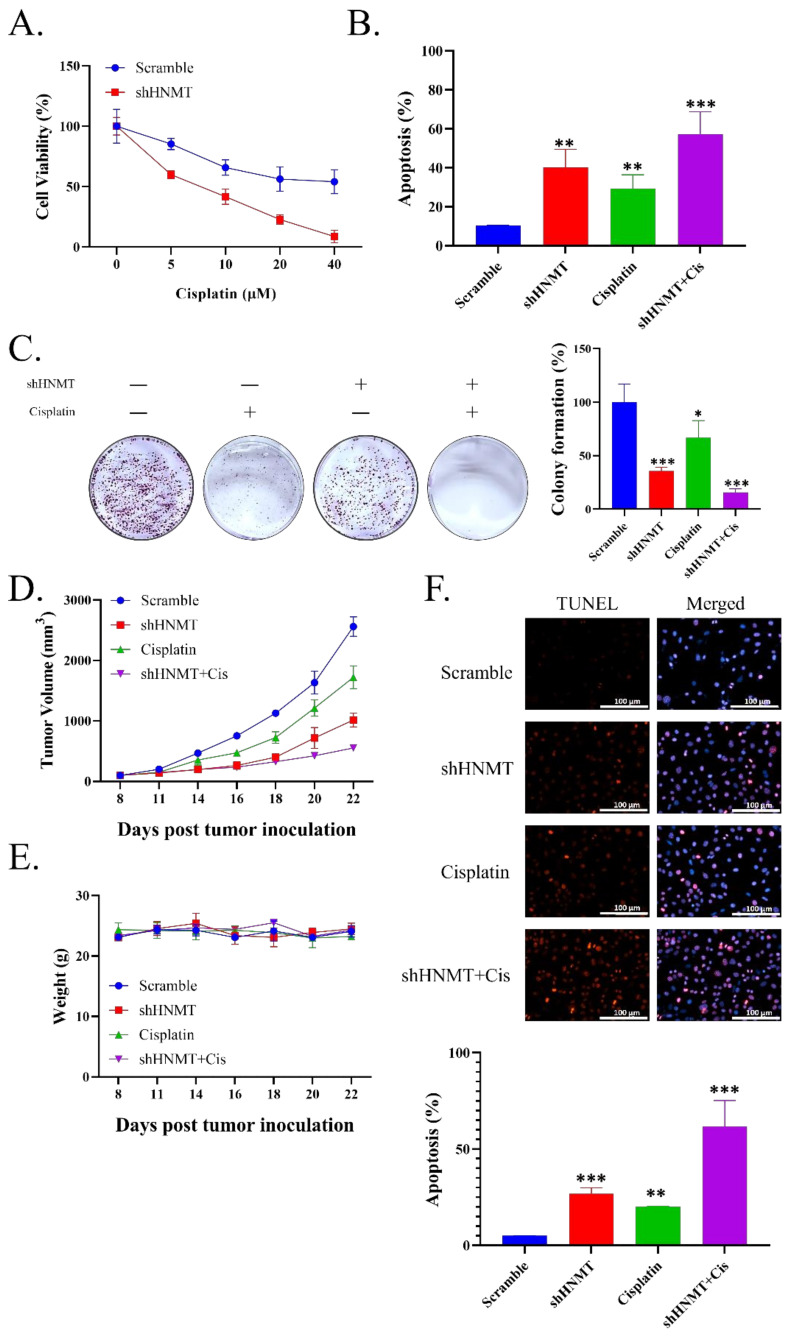
Association of HNMT with cisplatin resistance in lung cancer. (**A**) Cell viability (SRB) assay performed on A-549 cell line with or without shHNMT transfection followed closely by treatment with various dosages of cisplatin for 48 h. (**B**) Annexin V detection of apoptosis in H441 cells transfected with shScramble or shHNMT and incubated with or without 4 µM cisplatin for 48 h. Live cells (lower left square), dead cells (upper left square), cells in early apoptosis (lower right square), and late apoptotic cells (upper right square). (**C**) Colony formation assay of H441 cells transfected with either shScramble or shHNMT and incubated with or without 1 µM cisplatin for 10 days. Cells were then stained with crystal violet solution. (**D**) Tumor volume of mice xenografted with H441 cells with either shScramble or shHNMT was evaluated every 3 days after day 8 of tumor implantation to evaluate cisplatin treatment effectiveness. (**E**) Mouse body weight has been evaluated every 3 days. (**F**) Representative images of the TUNEL assay (red fluorescence for apoptotic cells and blue fluorescence for cell nuclei have been detected using a fluorescence microscope; magnification, 400×). The apoptosis rate is depicted in the bottom panel. ** p* < 0.05, *** p* < 0.01, **** p* < 0.001. Scale bar: 100 μm.

**Figure 7 ijms-23-01663-f007:**
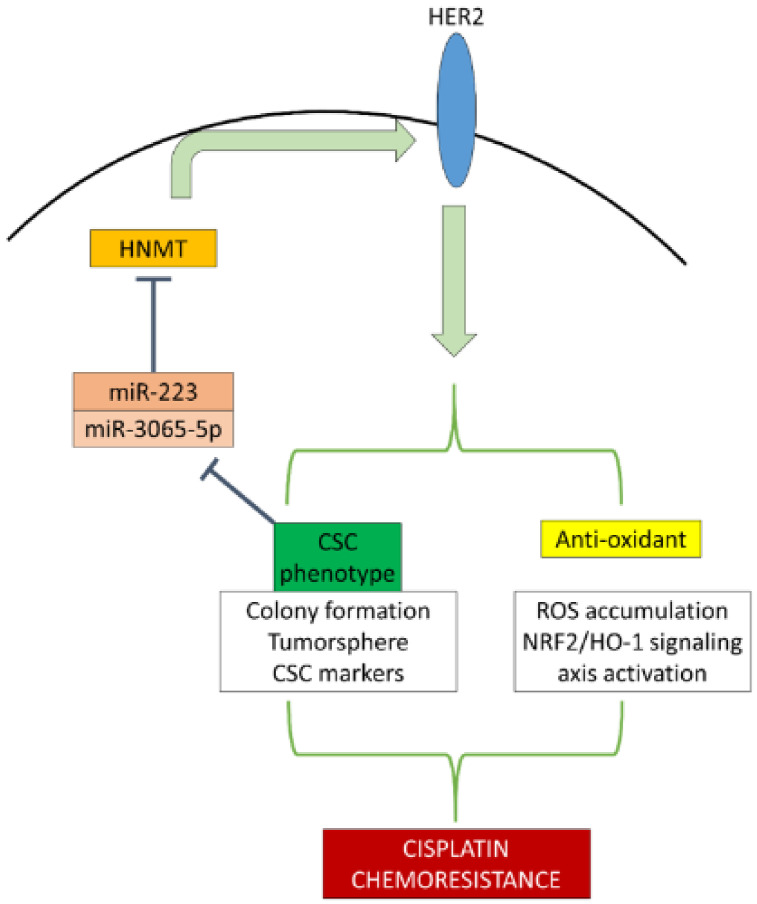
Graphical summary. HNMT interacts with HER2 to induce CSCs phenotype development and enhance NSCLC antioxidant properties, which in turn attenuate miR-223/3065 transcription that loops back to increased HNMT expression. HNMT/HER2-induced CSCs and antioxidant enhancement subsequently resulted in cisplatin chemoresistance in NSCLCs.

**Table 1 ijms-23-01663-t001:** The summary of clinicopathological features and distribution of HNMT status in 60 patients with NSCLC patients.

Clinicopathological Variables	N = 60	HNMT		x^2^	*p*-Value
		High Expression	Low Expression		
**Age, years**					
≤60	**29**	11	18	1.133	0.287
>60	**31**	16	15		
**Gender**					
Male	**37**	18	19	0.519	0.471
Female	**23**	9	14		
**Differentiation**					
Well/Moderately	**38**	13	25	4.875	0.027
Poor	**22**	14	8		
**Tumor Size (cm)**					
≤5	**39**	12	27	9.118	0.003
>5	**21**	15	6		
**Lymph node metastasis**					
N0	**36**	11	25	7.587	0.006
N1-N2	**24**	16	8		
**Clinical Stage**					
Early-stage (I-II)	**30**	7	23	11.38	<0.001
Late-stage (III-IV)	**30**	20	10		
**Subtypes**					
Adeno	**37**	20	17	3.197	0.074
Squamous	**23**	7	16		
**HER2 Q score**					
Low	**35**	11	24	6.251	0.012
High	**25**	16	9		

**Table 2 ijms-23-01663-t002:** Univariate and multivariate analysis of HNMT expression in the NSCLC cohort.

Parameter	Univariate				Multivariate			
	HR	95% CI		*p*	HR	95% CI		*p*
Gender (Female vs. male)	0.679	0.274	1.683	0.403	0.594	0.179	1.971	0.395
age_60 (<60 vs. >60)	0.857	0.360	2.037	0.726	0.624	0.211	1.840	0.392
tumor_50 (<50 vs. >50)	0.721	0.297	1.753	0.471	0.296	0.054	1.632	0.162
LN_12 (1,2 vs. 0)	0.675	0.278	1.637	0.384	0.427	0.125	1.458	0.174
Subtype_c(squa vs. adeno)	0.999	0.413	2.412	0.998	0.738	0.224	2.431	0.618
stage (late vs. early)	1.140	0.479	2.711	0.768	1.011	0.316	3.235	0.986
Differentiation_c(well, moderately vs. poor)	0.952	0.400	2.267	0.911	0.472	0.099	2.255	0.347
HNMT (low vs. high)	0.294	0.108	0.806	0.017	0.152	0.047	0.495	0.002
HER2(low vs. high)	1.061	0.442	2.546	0.894	1.346	0.408	4.445	0.626

## Data Availability

The datasets used and analyzed in the current study are publicly accessible, as indicated in the manuscript.

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
