# Peer review of "HNMT Upregulation Induces Cancer Stem Cell Formation and Confers Protection against Oxidative Stress through Interaction with HER2 in Non-Small-Cell Lung Cancer"

_ijms, 2022, doi:10.3390/ijms23031663_

Round 1

Reviewer 1 Report

I found your research very interesting and I learned a lot from it.
Your research was convincing, with a wealth of clinical and basic experimental data, and many additional experiments to support your hypothesis.
I look forward to the further development of your research in this field.

Author Response

Point-by-point responses to reviewer’s comments:

:::::::::::::::::::::::::::::::::::::::::::::::::::::::::::::::::::::::::::::::::::::::::::::::::::::::::::::::::::::::::::::::::::::::

R1: I found your research very interesting, and I learned a lot from it. Your research was convincing, with a wealth of clinical and basic experimental data, and many additional experiments to support your hypothesis. I look forward to the further development of your research in this field.

A1: We would like to again thank the Reviewer for the thorough reading of our manuscript as well as the valuable comments and appraisal. It will help us to grow more confidently in the current area of the research field.  

:::::::::::::::::::::::::::::::::::::::::::::::::::::::::::::::::::::::::::::::::::::::::::::::::::::::::::::::::::::::::::::::::::::::

Reviewer 2 Report

In the current manuscript the authors describe their findings regarding the role of HNMT in interaction with HER2 signaling, stemness and prognosis in NSCLC. The paper is well-written, the experimental approach exhaustive and sound, and the overall presentation and discussion logical and coherent.

A point of concern is the fact that when using the Prognoscan database for correlation with clinical progression the findings do not seem to have been corrected for other parameters. Is HNMT also prognostic after multivariate analysis (e.g. taking TNM stage into account)? Also, is there a difference between NSCLC subtypes (e.g. AdCa vs. SqCC) which would go some way towards explaining the stratification? 

Author Response

Point-by-point responses to reviewer’s comments:

:::::::::::::::::::::::::::::::::::::::::::::::::::::::::::::::::::::::::::::::::::::::::::::::::::::::::::::::::::::::::::::::::::::::

Dear Reviewer,

Co-authors and I very much appreciated the encouraging, critical, and constructive comments on this manuscript by the reviewer. The comments have been very thorough and useful in improving the manuscript. We strongly believe that the comments and suggestions have increased the scientific value of the revised manuscript by many folds. We have taken them fully into account in revision. We are submitting the corrected manuscript with the suggestion incorporated in the manuscript. The manuscript has been revised as per the comments given by the reviewer, and our responses to all the comments are as follows:

:::::::::::::::::::::::::::::::::::::::::::::::::::::::::::::::::::::::::::::::::::::::::::::::::::::::::::::::::::::::::::::::::::::::

Reviewer #2:

Q1: In the current manuscript the authors describe their findings regarding the role of HNMT in interaction with HER2 signaling, stemness and prognosis in NSCLC. The paper is well-written, the experimental approach exhaustive and sound, and the overall presentation and discussion logical and coherent.

A1: We would like to again thank the Reviewer for the thorough reading of our manuscript as well as the valuable comments and appraisal.

:::::::::::::::::::::::::::::::::::::::::::::::::::::::::::::::::::::::::::::::::::::::::::::::::::::::::::::::::::::::::::::::::::::::

Q2: A point of concern is the fact that when using the Prognoscan database for correlation with clinical progression the findings do not seem to have been corrected for other parameters. Is HNMT also prognostic after multivariate analysis (e.g. taking TNM stage into account)? Also, is there a difference between NSCLC subtypes (e.g. AdCa vs. SqCC) which would go some way towards explaining the stratification?

A2: We agree with reviewers suggestions, in this revised manuscript we have incorporated the clinicopathological information of the NSCLC-SHH-patients cohort with the information of the status of HNMT and other factors associated with NSCLC, to answer the key point raised, together with the uni- and multivariate cox-proportional analysis of Univariate and multivariate analysis of HNMT expression in the NSCLC cohort. Kindly refer to the result section 2.1. of our edited manuscript, on page 3, lines 120-132, and below the text.

Furthermore, we validated the standard to compare the clinicopathologic parameters, as shown in Table 1 (below), we found that tumor differentiation, tumor size, lymph node metastasis (LMN) status, and pathological clinical-stage remained significantly different between HNMT high and low expression group. Additionally, the cell type was also a significant factor, with HNMT high expression is less frequent in NSCLC.

Subsequent univariate (UA) and multivariate analysis (MA), revealed that, ex-pression of HNMT and Her2, age, gender, tumor size, LNM, clinical stage, subtypes to evaluate significance of each variable upon the NSCLC-specific survival. Overexpres-sion of HNMT (hazard ratio 0.294, 95% confidence interval: 0.108 to 0.806, p<0.01 for UA and hazard ratio 0.152, 95% confidence interval: 0.047 to 0.495; p<0.002 for MA, Table 2) could be considered as independent prognostic factors in NSCLC patients. Taken together, these results imply that HNMT is associated with HER2 expression and is a potential therapeutic target in NSCLC.

:::::::::::::::::::::::::::::::::::::::::::::::::::::::::::::::::::::::::::::::::::::::::::::::::::::::::::::::::::::::::::::::::::::::

Table 1 Patient clinicopathological status of the NSCLC cohort.

Clinicopathological Variables

HNMT

N=60

High

Low

x2

p-value

expression

expression

Age, years

≤60

29

11

18

1.133

0.287

>60

31

16

15

Gender

Male

37

18

19

0.519

0.471

Female

23

9

14

Differentiation

Well/Moderately

38

13

25

4.875

0.027

Poor

22

14

8

Tumor Size (cm)

≤5

39

12

27

9.118

0.003

>5

21

15

6

Lymph node metastasis

N0

36

11

25

7.587

0.006

N1-N2

24

16

8

Clinical Stage

Early-stage (I-II)

30

7

23

11.38

<0.001

Late stage (III-IV)

30

20

10

Subtypes

Adeno

37

20

17

3.197

0.074

Squamous

23

7

16

HER2 Q score

Low

35

11

24

6.251

0.012

High

25

16

9

Table 2. Univariate and multivariate analysis of HNMT expression in the NSCLC cohort.

Univariate

Multivariate

Parameter

HR

95 % CI

p

HR

95 % CI

p

Gender (Female vs male)

0.679

0.274

1.683

0.403

0.594

0.179

1.971

0.395

age_60 (<60 vs >60)

0.857

0.360

2.037

0.726

0.624

0.211

1.840

0.392

tumor_50 (<50 vs >50)

0.721

0.297

1.753

0.471

0.296

0.054

1.632

0.162

LN_12 (1,2 vs 0)

0.675

0.278

1.637

0.384

0.427

0.125

1.458

0.174

Subtype_c(squa vs adeno)

0.999

0.413

2.412

0.998

0.738

0.224

2.431

0.618

stage (late vs early)

1.140

0.479

2.711

0.768

1.011

0.316

3.235

0.986

Differentiation_c(well, moderately vs poor)

0.952

0.400

2.267

0.911

0.472

0.099

2.255

0.347

HNMT (low vs high)

0.294

0.108

0.806

0.017

0.152

0.047

0.495

0.002

HER2(low vs high)

1.061

0.442

2.546

0.894

1.346

0.408

4.445

0.626
